# TOKENSWAP: A Lightweight Method to Disrupt Memorized Sequences in LLMs

**Parjanya Prajakta Prashant** *
UC San Diego

**Kaustubh Ponkshe** *
EPFL

**Babak Salimi**
UC San Diego

## Abstract

As language models scale, their performance improves dramatically across a wide range of tasks, but so does their tendency to memorize and regurgitate parts of their training data verbatim. This tradeoff poses serious legal, ethical, and safety concerns, especially in real-world deployments. Existing mitigation techniques, such as differential privacy or model unlearning, often require retraining or access to internal weights making them impractical for most users. In this work, we introduce TOKENSWAP, a lightweight, post-hoc defense designed for realistic settings where the user can only access token-level outputs. Our key insight is that while large models are necessary for high task performance, small models (e.g., DistilGPT-2) are often sufficient to assign fluent, grammatically plausible probabilities to common function words - and crucially, they memorize far less. By selectively swapping token probabilities between models, TOKENSWAP preserves the capabilities of large models while reducing their propensity for verbatim reproduction. Evaluations on Pythia-6.9B and Llama-3-8B show up to a $10\times$ drop in exact memorization with negligible task degradation. Our method offers a practical, accessible solution for mitigating memorized generation in deployed LLMs. Code is available at `https://github.com/parjanya20/verbatim-llm`.

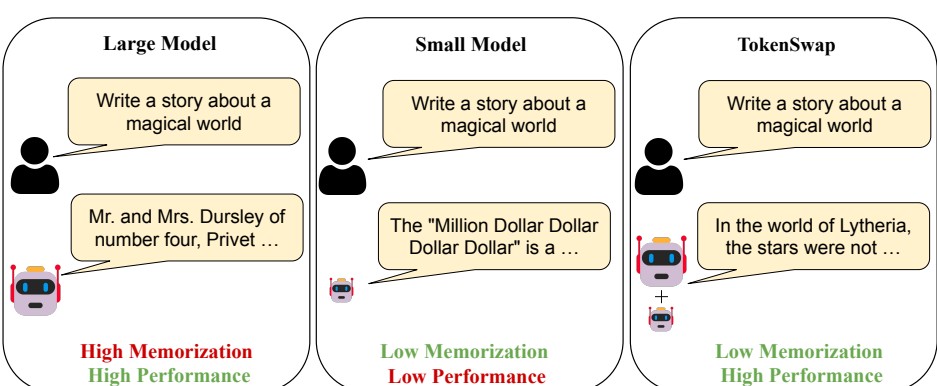

Figure 1: TOKENSWAP combines the strengths of large and small language models. Large models achieve high performance but exhibit high memorization. Small models have low memorization but poor performance (generating incoherent text). TOKENSWAP achieves both low memorization and high performance by selectively swapping token probabilities, generating novel, fluent text.

*Equal Contribution; Correspondence to Parjanya Prajakta Prashant <pprashant@ucsd.edu>, Kaustubh Ponkshe <kaustubh.ponkshe@epfl.ch>

39th Conference on Neural Information Processing Systems (NeurIPS 2025).

# 1 Introduction

Large language models (LLMs) such as GPT–4, GEMINI, and LLAMA have demonstrated strong performance across a wide range of tasks, from natural language understanding to complex reasoning [3, 65, 24]. These capabilities are driven by their massive parameter counts and extensive training corpora, enabling human-level fluency and impressive reasoning across domains. Often referred to as emergent properties, such abilities arise directly from scale, with well-established scaling laws predicting performance gains. However, increased scale also introduces a critical drawback: the tendency of LLMs to memorize and reproduce parts of their training data [15, 14, 11, 48].

One of the most pressing consequences of memorization is the verbatim or near-verbatim generation of training data [38, 66, 6]. Although memorization is an inherent property and not necessarily harmful, its consequence of verbatim generation leads to plagiarism and copyright violation. This behavior poses serious risks to both model providers and end-users. Providers may face legal challenges, including copyright infringement lawsuits [38, 29, 51], while users unknowingly risk legal liability by reproducing protected content. Crucially, the threat is not limited to exact substring matches: even approximate or near-verbatim outputs can constitute infringement, as evidenced by lawsuits like the *New York Times* case against OpenAI for near-verbatim content generation [28]. Moreover, verbatim generation can occur even in benign scenarios where users have no adversarial intent to extract training data [4, 21]. Users may unknowingly generate copyrighted content during routine interactions, exposing themselves to unintended legal risks [22].

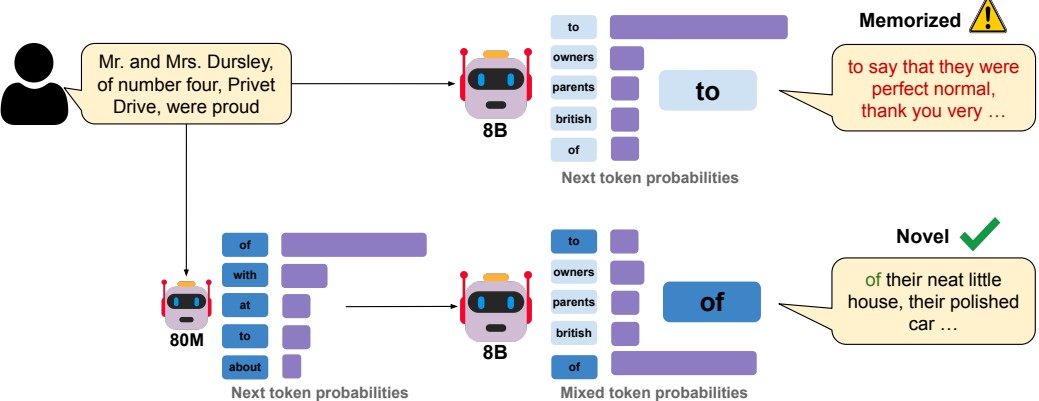

Figure 2: Overview of TOKENSWAP. Our approach replaces token probabilities of high-frequency "grammar-based" tokens with those from a small auxiliary language model. This mitigates memorized generation while maintaining fluency and model performance. The top path shows standard LLM generation, while the bottom path demonstrates how TOKENSWAP alters token selection to disrupt memorization and produce novel text.

We consider the perspective of a typical user of commerical LLMs such as GPT-4 [3], GEMINI [65], LLAMA [24], and DEEPSEEK [42]. These models do not share their training data and many do not make their weights publicly available. Even in cases where weights are openly shared, hosting a production-grade LLM requires substantial memory resources, rendering it impractical for the average user. Consequently, it is reasonable to assume that most users can only interact with these models through APIs hosted on external servers, with access limited to model outputs such as token-level logits. Despite these practical constraints, to our knowledge, none of the existing methods, whether designed to prevent memorization or mitigate verbatim output, can effectively operate under such limited access conditions.

**Existing methods require access to training data and/or model weights** Approaches to address memorization are broadly categorized into pre-training and post-training interventions. Pre-training methods include deduplication [37], differential privacy (DP) [2], and selective token exclusion during training [31]. While these approaches can reduce memorization, they often incur substantial computational costs and degrade model performance [7]. Post-training interventions focus on un-learning techniques that attempt to modify specific neurons and weights or utilize finetuning methods

to prevent models from generating memorized content [44, 56, 18, 54]. However, these methods remain susceptible to training data extraction [60], often impair general model capabilities [34], and can lead to unintended forgetting of critical aspects such as safety guardrails [67]. This challenge is further complicated by theoretical findings suggesting that some degree of memorization may be inherent to achieving generalization in learning algorithms [8].

In contrast, another line of work focuses on preventing the generation of memorized content at inference time without modifying model weights. These approaches include blocking exact matches to training data [35] or combining logits from multiple models trained on disjoint datasets [1]. However, these methods too require access to training data or multiple LLMs trained on strictly disjoint datasets. Table 1 summarizes the various approaches and assumptions under which they operate (see Appendix A for a comprehensive review).

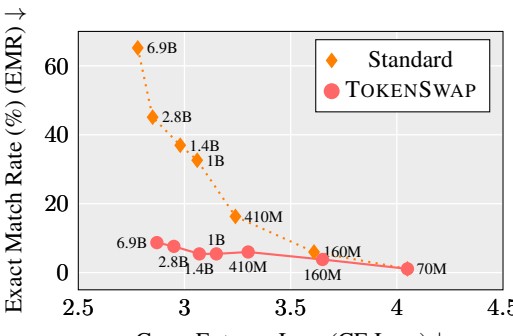

Figure 3: Memorization (EMR) vs Performance (CE Loss) across different model sizes. Larger, more capable models exhibit higher memorization. TOKENSWAP, with Pythia-70M as the auxiliary model, achieves low memorization rates while maintaining competitive performance. Details in Section 4.2 and Section 5.

**Memorization scales with size** The propensity to reproduce training data consistently increases with the size of the language model [14, 11]. Since model performance generally scales positively with size, users are forced into a trade-off between obtaining high performance and mitigating memorized generations. Figure 3 demonstrates this relationship using a series of Pythia models, showing the trade-off between memorization and cross-entropy loss.

In this work, we present TOKENSWAP, an inference-time method that significantly alleviates this tradeoff by combining large model performance with small model memorization (Figure 3). TO-KENSWAP selectively replaces the probabilities of a subset of common grammar tokens (e.g., "the", "of", "and") of the large main model with those from a small auxiliary model. This technique disrupts the verbatim generation by breaking the high-probability paths that lead to verbatim reproduction. This disruption has a cascading effect: once one token deviates from the memorized sequence, all subsequent predictions are conditioned on this altered context, further preventing reproduction. Importantly, since small models reliably approximate probabilities for common grammatical tokens, TOKENSWAP preserves the large model's performance. For auxiliary models of size much smaller than the main model, this provides a verbatim memorization mitigation method which requires access *neither to the training data nor the model weights*. Since we treat the effect of memorization, and not the cause itself, our method is able to reduce verbatim generation at inference time.

We extensively evaluate TOKENSWAP through both controlled experiments and real-world deployments. In controlled fine-tuning experiments (Section 4.1), TOKENSWAP achieves a 50-800× reduction in verbatim generation compared to undefended models. Evaluations on commercial-grade models such as Pythia-6.9b and Llama-3-8b (Section 4.2) demonstrate reductions in verbatim generation by upto 10×, without compromising downstream task performance. Furthermore, comparisons with Goldfish [31] show that TOKENSWAP matches or surpasses the effectiveness of state-of-the-art pre-training methods (Section 4.3).

# 2 Preliminaries

## 2.1 Language Models: Notation and Setup

We consider auto-regressive language models that model the log-probability of a token conditioned on all previous tokens in a sequence. They operate over a vocabulary $\mathcal{V} = \{v_1, \ldots, v_{|\mathcal{V}|}\}$ of typically $|\mathcal{V}| \approx 10^5 - 10^6$ tokens. Given an input prompt $(x_{-l_p}, \ldots, x_{-1}) \in \mathcal{V}^{l_p}$ of length $l_p$ followed by a response sequence $(x_0, \ldots, x_{l-1}) \in \mathcal{V}^l$ of length $l$, an auto-regressive language model parametrizes the joint probability by factorizing over conditional probabilities:

Table 1: Comparison of TOKENSWAP with existing methods based on their assumptions. TO-KENSWAP uniquely avoids requiring access to model weights or the copyrighted training corpus. While it employs an auxiliary model, the memory overhead is small $\approx 1\%$ due to the small size of the auxiliary model. PT: pre-training, UL: unlearning, FT: fine-tuning, Inf: inference-time.

| | Model Access | Copyrighted Corpus Access | Inference Overhead |
|---|---|---|---|
| Deduplication [37] PT | weights | ✓ | ✗ |
| Goldfish [31] PT | weights | ✓ | ✗ |
| Balanced subnet [56] UL | weights | ✓ | ✗ |
| Obliviate [54] FT | weights | ✓ | ✗ |
| MemFree [35] Inf | logits | ✓ | efficient querying |
| CP-Fuse [1] Inf | logits | ✓ | twice of standard generation |
| TOKENSWAP Inf | logits | ✗ | small auxiliary model |

$$p(x_0, \ldots, x_l | x_{-l_p}, \ldots, x_{-1}) \;=\; \prod_{i=0}^{l} p(x_i \,|\, x_{<i}), \tag{1}$$

For each position $i$, the model outputs a distribution $\mathbf{p}_i[v]$ over $\mathcal{V}$, where $\mathbf{p}_i[v] = p(x_i = v | x_{<i})$.

Since language models are trained to maximize the likelihood of observed sequences, they tend to assign high probabilities to tokens that frequently follow specific prefixes during training. This increases the risk of memorization and *verbatim reproduction* of training data.

## 2.2 Extractable Memorization

Memorization in language models can manifest in various ways, but a practically relevant and widely adopted framework is *extractable memorization* [15, 14]. Carlini et al. [15] demonstrate that models can be induced to regurgitate training sequences when prompted with prefixes from their training data. The following definition formalizes this concept:

**Definition 1** (Extractable Memorization). A sequence $x = (x_0, \ldots, x_{l-1})$ of length $l$ is considered *extractable with $l_p$ tokens of context* from a language model $p$ if there exists a prefix $x_- = (x_{-l_p}, \ldots, x_{-1})$ of length $l_p$ such that $[\, x_- \,\|\, x \,]$ appears in the training data of $p$, and $p$ reproduces $x$ via greedy decoding.
Formally, for each $i \in \{\, 0, \ldots, l-1 \,\}$:

$$x_i \;=\; \arg\max_{x' \in \mathcal{V}} \, p\big(x' \,|\, x_{<i}, x_-\big).$$

This definition is practically useful because: (1) it aligns with real-world risks of copyright and memorized generation [47, 38], (2) it provides a concrete, testable condition that can be evaluated on real models, and (3) it extends to models of different sizes, capturing the well-documented trend that larger models memorize more data [14, 11]. This scaling behavior is important in motivating our methodology in Section 3.

## 3 Methodology

As discussed earlier, small language models (e.g., DistilGPT-2, Pythia-70M) have lower propensity to reproduce training data compared to large models (e.g., Llama3, GPT-4). We introduce TOKENSWAP, a lightweight, post-hoc method that combines the strengths of both model scales: large-model performance with small-model memorization. During inference, TOKENSWAP replaces the probabilities for selected tokens of a large model with those a small model.

**Algorithm** Let $\mathbf{p}^{\text{main}}$ and $\mathbf{p}^{\text{aux}}$ denote the probability distributions of the main and auxiliary models respectively, where $\mathbf{p}^{\text{main}}(x_t \,|\, x_{<t})$ and $\mathbf{p}^{\text{aux}}(x_t \,|\, x_{<t})$ represent their token probabilities conditioned on previous tokens. We assume the parameter count of the main model significantly exceeds that of the auxiliary model. Given these models, TOKENSWAP selectively replaces probabilities for a fixed subset of tokens $\mathcal{G} \subset \mathcal{V}$. The complete procedure is formalized in Algorithm 1.

At each position $i$, TOKENSWAP queries both $\mathbf{p}^{\text{main}}$ and $\mathbf{p}^{\text{aux}}$ to obtain probability distributions conditioned on the current context $x_{<i}$. For tokens in subset $\mathcal{G} \subset \mathcal{V}$, probabilities from the main model are replaced with scaled probabilities from the auxiliary model, with scaling factor $\alpha$ ensuring the final distribution $\mathbf{p}^{\text{final}}$ remains a valid distribution. This prevents reproduction of memorized sequences: if any token $x_i$ in a memorized sequence belongs to $\mathcal{G}$, its probability under $\mathbf{p}^{\text{final}}$ is determined by the auxiliary model. Since the auxiliary model memorizes less, this disrupts the chain of conditional probabilities required for verbatim generation of most sequences. Importantly, for tokens $v \notin \mathcal{G}$, their probabilities remain unchanged, i.e., $\mathbf{p}_i^{\text{final}}[v] = \mathbf{p}_i^{\text{main}}[v]$.

---

**Algorithm 1** TOKENSWAP

---

**Require:** Main model $\mathbf{p}^{\text{main}}$, auxiliary model $\mathbf{p}^{\text{aux}}$, token subset $\mathcal{G}$, prompt $x_{<0}$
1: **for** $i = 0, 1, \ldots$ **do**
2: $\quad \mathbf{p}_i^{\text{main}} \leftarrow \mathbf{p}^{\text{main}}(\cdot | x_{<i})$ {Get main model probabilities}
3: $\quad \mathbf{p}_i^{\text{aux}} \leftarrow \mathbf{p}^{\text{aux}}(\cdot | x_{<i})$ {Get auxiliary model probabilities}
4: $\quad \alpha \leftarrow \frac{\sum_{v \in \mathcal{G}} \mathbf{p}_i^{\text{main}}[v]}{\sum_{v \in \mathcal{G}} \mathbf{p}_i^{\text{aux}}[v]}$ {Compute normalization}
5: $\quad$ **for** $v \in \mathcal{V}$ **do**
6: $\quad\quad \mathbf{p}_i^{\text{final}}[v] \leftarrow \begin{cases} \mathbf{p}_i^{\text{main}}[v], & \text{if } v \notin \mathcal{G} \\ \alpha \cdot \mathbf{p}_i^{\text{aux}}[v], & \text{if } v \in \mathcal{G} \end{cases}$
7: $\quad$ **end for**
8: $\quad x_i \sim \mathbf{p}_i^{\text{final}}$ {Sample next token}
9: **end for**

---

**Selecting $\mathcal{G}$ for Effective Memorization Disruption**   The choice of $\mathcal{G}$ affects both memorization and model performance. By modifying token probabilities, TOKENSWAP disrupts memorized sequences while preserving fluency. However, not all tokens are equally effective for this purpose. $\mathcal{G}$ should consist of tokens that frequently appear in memorized text, as replacing their probabilities reduces the likelihood of exact reproduction. At the same time, modifying inappropriate tokens can degrade model performance, especially for specialized tasks. For instance, if $\mathcal{G}$ includes numeric tokens, mathematical reasoning may degrade. Therefore, $\mathcal{G}$ should satisfy two key criteria. First, it must contain frequently occurring tokens. Second, it should avoid tokens where probability replacement impacts the model's capabilities.

Empirical studies suggest that small models correctly approximate the probabilities of high-frequency function words while diverging more on rare or domain-specific terms [52]. Additionally, small language models ($\approx 100M$) can generate coherent and grammatically correct text [25]. Based on these insights, we construct $\mathcal{G}$ from grammar-based high-frequency tokens (e.g. - 'the', 'in'). Further, since $\mathcal{G}$ consists of high-frequency words, there exists a natural one-to-one mapping between tokens even when $\mathbf{p}^{\text{main}}$ and $\mathbf{p}^{\text{aux}}$ use different tokenizers and vocabularies. While this approach is well-suited for natural language, structured domains such as code may require domain-specific adaptations. Additional details on the construction of $\mathcal{G}$ are provided in Appendix C.2 and C.3.

## 4   Experiments

In this section, we demonstrate the effectiveness of TOKENSWAP, in both controlled and real-world settings. Our experiments evaluate TOKENSWAP along two dimensions:

- The method's efficacy in preventing exact *and approximate* reproduction of training data.

- The impact on model performance across common-sense reasoning, language and fluency.

We evaluate TOKENSWAP across three settings to demonstrate its effectiveness. In Section 4.1, we deliberately induce memorization through extensive fine-tuning on small datasets to stress-test our defense. Section 4.2 evaluates TOKENSWAP on production-grade models including Pythia-6.9B and Llama-3-8B. Finally, in Section 4.3, we compare against Goldfish [31], a pre-training method specifically designed to reduce memorization, showing that our post-hoc approach achieves comparable results without requiring model retraining.

## 4.1 Extreme Memorization

In order to rigorously evaluate TOKENSWAP, we create an extreme test case by deliberately inducing memorization through extensive fine-tuning. While TOKENSWAP can be applied to real-world models directly, our baselines require specific experimental conditions for comparison. Similar extreme test cases have been generated to evaluate memorization in prior work [31, 1]. Following Abad et al. [1], we fine-tune a Llama-3.2-3B model [24] on 2,000-sequence subsets from two datasets: MathAbstracts [69] and WritingStories [27]. We train for 50 epochs to deliberately amplify memorization beyond typical levels.

**Memorization Metrics** Our analysis employs both exact and approximate memorization and performance metrics to ensure a comprehensive assessment. Exact memorization is measured through *Matching Length (ML)*, which the number of verbatim characters or tokens generated before first deviation, and *Exact Matching Rate (EMR)*, which computes the fraction of sequences reproduced verbatim. To capture partial memorization, we use the *ROUGE-L* score, which identifies the longest common non-contiguous subsequence and gives a score between 0 and 1, and the *Normalized Levenshtein Distance*, which quantifies the minimum number of edits needed to transform generated text into the original sequence. Lower scores indicate reduced memorization for Matching Length, Exact Matching Rate, and ROUGE-L. Higher scores are better for Normalized Levenshtein Distance. These metrics are widely used to evaluate verbatim and approximately verbatim generation [38, 31, 1].

**Performance Metrics** Since our setup intentionally induces extreme memorization, standard performance metrics are not meaningful. Nonetheless, we report cross-entropy loss on a held-out validation set in Appendix B.2.

**Setup and Inference-time Baselines** We compare against the two inference-time baselines: CP-Fuse [1], which samples from weighted combinations of models trained on disjoint datasets, and MemFree [35], which blocks exact $n$-gram matches to the training data. Standard refers to greedy decoding without any memorization mitigation. Both baselines rely on unrealistic assumptions-MemFree requires access to the training data, while CP-Fuse assumes access to two separately trained models on disjoint corpora. To assess CP-Fuse under more realistic conditions, we evaluate two variants: CP-FUSE HALF, with perfectly disjoint sets of 1,000 sequences each, and CP-FUSE MIXTURE, with 1,500 sequences per model and 500 overlapping. For TOKENSWAP, we employ DistilGPT-2 (80M) [57] as $\mathbf{p}^{\text{aux}}$. We construct $\mathcal{G}$ with $|\mathcal{G}| = 110$ tokens using high-frequency 'grammar-based' words. Additional details on $\mathcal{G}$ are provided in Appendix C.2. For all experiments and methods, a prefix of 20 tokens is used and the next 128 tokens are greedily sampled.

Table 2: Comparison of memorization mitigation methods for WritingPrompts and MathAbstracts datasets. Memorization metrics: Matching Length (ML), Exact Match Rate (EMR), Normalized Levenshtein Distance (Levenshtein), ROUGE-L. Models used: Finetuned Llama-3.2-3B.

| Method | WritingPrompts | | | | MathAbstracts | | | |
|---|---|---|---|---|---|---|---|---|
| | ML↓ | EMR↓ | ROUGE-L↓ | Lev.↑ | ML↓ | EMR↓ | ROUGE-L↓ | Lev.↑ |
| Standard | 464.0 | 83.4 | 0.89 | 0.10 | 450.4 | 93.6 | 0.98 | 0.03 |
| MemFree | 17.4 | 0.0 | 0.29 | 0.63 | 6.7 | 0.0 | 0.44 | 0.55 |
| CP-Fuse-mix | 280.3 | 49.2 | 0.58 | 0.37 | 233.7 | 47.1 | 0.62 | 0.36 |
| CP-Fuse-half | 12.5 | 0.0 | 0.17 | 0.73 | 15.3 | 0.1 | 0.26 | 0.71 |
| TOKENSWAP | **19.7** | **0.1** | **0.19** | **0.71** | **53.0** | **1.8** | **0.38** | **0.60** |

**Results** Table 2 demonstrates TOKENSWAP's effectiveness in reducing memorization across both datasets. For WritingPrompts, TOKENSWAP reduces EMR by 800x (from 83.4% to 0.1%) and ROUGE-L by 4.6x (from 0.89 to 0.19) compared to standard generation. On MathAbstracts, EMR decreases by 50x (from 93.6% to 1.8%) and ROUGE-L by 2.6x (from 0.98 to 0.38). CP-Fuse-half achieves slightly better results but requires disjoint training sets, while CP-Fuse-mix performs significantly worse due to dataset overlap. MemFree achieves the lowest scores on the exact memorization metrics (Exact Matching Rate and Matching Length) but performs poorly on approximate memorization metrics (ROUGE-L and Levenshtein). This shows that, while MemFree prevents verbatim generation, it still allows high levels of near-verbatim generation. The performance gap between

**Prefix:** Magic – once a real and potent force but as the world population

| Original Suffix | Standard Generation | TOKENSWAP Generation |
|---|---|---|
| grew from millions to billions the shared mana per person is now negligible. A group of astronauts helplessly watching the Earth perish experience... | grew from millions to billions the shared mana per person is now negligible. A group of astronauts helplessly watching the Earth perish experience... | grew and the number of wizards and witches declined, the world began to suffer. Now the world suffers from a lack of magic, and the government is tasked with... |

Figure 4: Comparison of text generation methods. Red text indicates memorized content. Standard generation reproduces the entire suffix verbatim, while TOKENSWAP generates novel content.

WritingPrompts (EMR: 0.1%) and MathAbstracts (EMR: 1.8%) aligns with our intuition - $\mathcal{G}$ was designed focusing on natural language tasks. Nevertheless, TOKENSWAP achieves substantial memorization reduction for both domains. To complement our quantitative results, *we provide qualitative examples of generations from the WritingPrompts dataset* in Figure 4 and Appendix E.

## 4.2 Memorization in the wild

In this section, we demonstrate the efficacy of our approach on production-grade models. We assess the effectiveness of TOKENSWAP on two pre-trained models: Pythia-6.9B [12] and Llama-3-8B [24].

**Pile-Memorized Dataset** For Pythia-6.9B, we evaluate on memorized sequences identified by Chang et al. [16] from the Pile dataset, consisting of 32-token prefixes and 48-token suffixes. After filtering to retain only natural language content (excluding code, URLs, etc.), we obtain 184 evaluation examples.

**LeetCode Dataset** For Llama-3-8B, following previous work demonstrating LeetCode problem memorization [38], we evaluate on 1,825 LeetCode problem statements [30]. *These problem statements are written in natural language.* Since the exact format of LeetCode problems in Llama's training data is unknown, we remove punctuation while calculating the memorization metrics. Additionally, instead of exact match rate, we use ROUGE-L $> 0.8$ as our threshold for identifying memorized content. Prefix length of 20 tokens is used and the next 100 tokens are sampled.

**Evaluation Setup** We face two key limitations when comparing TOKENSWAP with existing baselines. CP-Fuse requires models trained on disjoint datasets, but verifying this is difficult since most LLMs do not release training data. Even when available, disjoint datasets are unlikely given that most models train on overlapping web corpora like Common Crawl. Additionally, CP-Fuse requires identical tokenizers, limiting comparisons to models within the same family. Similarly, we cannot evaluate against MemFree due to unavailable training data (LLaMA) or prohibitively large datasets (Pythia uses the 800GB Pile [12]). To ensure fair evaluation for CP-Fuse, we paired each model with a smaller counterpart: Pythia-2.8B with Pythia-6.9B, and Llama-3.2-3B with Llama-3-8B. Using smaller models actually favors CP-Fuse since they memorize less. We avoid very small models ($<$100M) as CP-Fuse needs roughly equally capable models (see Appendix C.1.4). The setup for TOKENSWAP follows Section 4.1. For LeetCode evaluation, we use both DistilGPT-2 and SmolLM-135M [5] as auxiliary models. SmolLM is an instruction-tuned model, which enables evaluation on instruction-following tasks like MT-Bench where an instruct-capable auxiliary model is required. For memorization, we use the same metrics as Section 4.1.

**Performance Metrics** We evaluate two key aspects: task performance and generation quality. For task performance, we assess five-shot learning on multiple commonsense reasoning benchmarks: BoolQ [19], SIQA [58], PIQA [13], ARC-Challenge [20], ARC-Easy [20], OBQA [45], and Wino-Grande [55]. For generation quality, we report cross-entropy loss on samples from *Slimpajama* [61], which correlates with fluency [10] and has been used to evaluate prior memorization mitigation

work [31, 1]. We also evaluate on MT-Bench [70], which tests multi-turn conversation, instruction-following, and generation quality through realistic conversational scenarios. Note that MT-Bench and commonsense reasoning results are only reported for Llama-3-8B (LeetCode Dataset) since these require instruction-following capabilities not available in the base Pythia models.

Table 3: Comparison of mitigation methods for LeetCode and Pile-Memorized datasets. Memorization metrics: Matching Length (ML), Exact Match Rate (EMR), Normalized Levenshtein Distance (Levenshtein), ROUGE-L & Performance metrics: Cross Entropy Loss (CE Loss) on SlimPajama, MT-Bench with GPT-4 as a judge, Mean of scores on Commonsense Reasoning benchmarks. Models used: Llama-3-8B and Pythia-6.9B.

| **LeetCode Dataset (Llama)** | | | | | | | |
|---|---|---|---|---|---|---|---|
| Method | ML ↓ | ROUGE-L> 0.8 ↓ | ROUGE-L ↓ | Lev. ↑ | CE Loss ↓ | MT-Bench ↑ | Commonsense ↑ |
| Standard | 24.57 | 9.65 | 0.39 | 0.60 | 2.38 | 7.75 | 71.87 |
| CP-Fuse | 19.44 | 7.01 | 0.37 | 0.61 | 2.45 | 8.53 | 70.18 |
| TOKENSWAP [1] | **6.04** | **0.96** | **0.27** | **0.71** | 2.52 | - | **71.87** |
| TOKENSWAP [2] | 8.58 | 1.92 | 0.30 | 0.69 | 2.43 | 7.78 | - |

| **Pile-Memorized Dataset (Pythia)** | | | | | |
|---|---|---|---|---|---|
| Method | ML ↓ | EMR ↓ | ROUGE-L ↓ | Lev. ↑ | CE Loss ↓ |
| Standard | 151.6 | 65.22 | 0.80 | 0.18 | 2.80 |
| CP-Fuse | 97.05 | 29.35 | 0.62 | 0.35 | 2.81 |
| TOKENSWAP [1] | **35.10** | **5.98** | **0.38** | **0.56** | 2.88 |

[1]DistilGPT-2 as auxiliary model. [2]SmolLM-135M as auxiliary model.

**Results**    Table 3 demonstrates that TOKENSWAP substantially reduces memorization across both datasets compared to standard generation and CP-Fuse. Exact match rate decreases by over 10x compared to standard generation and 5-7x compared to CP-Fuse on both datasets. The average matching length shows similar improvements, reducing by 4-5x versus standard and 3-4x versus CP-Fuse. The consistent improvements in approximate memorization metrics (ROUGE-L and Levenshtein distance) demonstrate that TOKENSWAP robustly prevents verbatim generation rather than simply introducing small perturbations. CP-Fuse shows limited effectiveness in these real-world scenarios primarily because its core assumption of disjoint training datasets does not hold. Even when using different models, the inherent overlap in web-scale training corpora prevents CP-Fuse from effectively disrupting memorized sequences.

TOKENSWAP maintains task performance by selectively targeting only grammar-based tokens, leaving reasoning-critical content words unchanged. This preserves commonsense reasoning abilities, as shown by identical accuracy scores (71.87%) compared to standard generation. The method also maintains fluency, evidenced by minimal cross-entropy increases and nearly equal MT-Bench scores. While CP-Fuse achieves better conversational performance (8.53 vs 7.78), it fails to verbatim generation, making it unsuitable for the desired goal.

**Evaluation on OLMo-2-13B.**    To further validate TOKENSWAP on a fully open model with known training data, we evaluate TOKENSWAP on OLMo-2-13B [49]. The full training corpus contains 3T tokens, making exhaustive memorization search infeasible. We therefore focus on the Wikipedia subset of the training data. Results in Appendix B.1 show that TOKENSWAP eliminates exact verbatim generation (EMR=0) and substantially reduces approximate memorization.

### 4.3   Comparison with Pre-training Methods

While previous sections demonstrate that TOKENSWAP outperforms post-hoc baselines, we also compare with Goldfish [31], a pre-training approach that reduces memorization by excluding a fraction $1/k$ of tokens from loss computation during training. Since pre-training large models using this loss is expensive, we evaluate on pre-trained goldfish models from Hans et al. [31]. These models were trained on a subset of RedPajama [68] combined with 2000 Wikipedia sequences. To induce memorization, the Wikipedia sequences were duplicated 50 times during training. We compare against models trained with $k \in \{3, 4, 32\}$. For TOKENSWAP, we maintain the same experimental

setup from Section 4.1. Following Hans et al. [31], we use identical prefix and suffix lengths for extraction of memorized sequences.

Table 4: Comparison of TOKENSWAP with Goldfish [31] for $k \in \{3,4,32\}$. Memorization metrics: Matching Length (ML), Exact Match Rate (EMR), Normalized Levenshtein Distance (Levenshtein), ROUGE-L & Performance metrics: Cross Entropy Loss (CE Loss) on SlimPajama.

| Method | ML↓ | EMR↓ | ROUGE-L↓ | Levenshtein↑ | CE Loss↓ |
|---|---|---|---|---|---|
| Standard | 73.9 | 7.8 | 0.38 | 0.58 | 3.44 |
| Goldfish ($k$=3) | 12.7 | 0.0 | 0.23 | 0.72 | 3.54 |
| Goldfish ($k$=4) | 14.7 | 0.0 | 0.23 | 0.71 | 3.50 |
| Goldfish ($k$=32) | 58.1 | 2.5 | 0.35 | 0.60 | 3.44 |
| TOKENSWAP | **12.4** | **0.1** | **0.22** | **0.72** | **3.44** |
| TOKENSWAP + Goldfish ($k$=3) | **7.9** | **0.0** | **0.21** | **0.73** | **3.57** |

**Results** Table 4 shows TOKENSWAP achieves comparable or superior performance to Goldfish across all memorization metrics. Notably, TOKENSWAP obtains the best Matching Length, Rouge-L and Normalized Levenshtein distance scores while maintaining better cross-entropy than the Goldfish variants for $k = 3, 4$. The effectiveness of Goldfish varies with parameter $k$ - smaller values (more aggressive token exclusion) yield stronger memorization reduction but worse performance, as evidenced by higher cross-entropy. This illustrates a key advantage of TOKENSWAP: we achieve similar memorization reduction without requiring modified training or reduced training data tokens. Figure 5 (Appendix B.7) further supports this finding, showing nearly identical ROUGE-L score distributions between TOKENSWAP and Goldfish ($k$=3), indicating that our post-hoc approach matches the most aggressive pre-training variant. Furthermore, applying TOKENSWAP to Goldfish ($k = 3$) as the main model reduces memorization more than either method alone, demonstrating that our approach is orthogonal to pre-training methods and can enhance existing techniques.

## 5    Discussion and Limitations

In this section, we analyze TOKENSWAP's behavior across different settings. We first perform ablations on the auxiliary model choice and the size of $\mathcal{G}$. We then analyze TOKENSWAP across the Pythia model family to demonstrate significant improvements in the performance-memorization tradeoff. Finally, we discuss limitations and potential extensions of our method.

**Choice of the Auxiliary Model** In Section 4 we test TOKENSWAP with DistilGPT-2 as the auxiliary model. A natural question arises: *What auxiliary model should one choose and how does the size of the auxiliary model affect memorized generation?* To answer this, we use the SmolLM family [5] with three sizes (135M, 360M, 1.7B) and evaluate on both Pythia-6.9B (Pile-memorized dataset) and Llama-3-8B (LeetCode dataset). Detailed results are in Table 10 (Appendix B.6).

We observe a clear trend: smaller auxiliary models lead to less verbatim generation, confirming our hypothesis that TOKENSWAP's effectiveness stems from low memorization in auxiliary models. Importantly, auxiliary model size has minimal impact on performance. MT-Bench scores show negligible variation across auxiliary models—this is particularly significant since MT-Bench evaluates overall sequence generation quality, unlike cross-entropy loss which measures token-level accuracy. Therefore, any small model ($\approx 100M$) which can generate fluent text and predict grammar-based tokens well, such as DistilGPT-2 or SmolLM-135M, can be used effectively as an auxiliary model.

**Ablations on $\mathcal{G}$** The subset of tokens $\mathcal{G}$ is constructed by selecting grammar-based words from the top 500 most frequent English words, resulting in $|\mathcal{G}| = 110$ (details in Appendix C.2). To understand the impact of $\mathcal{G}$ size on memorization reduction, we ablate by constructing $\mathcal{G}$ from the top $k$ most frequent words for $k \in \{10, 50, 100, 500, 2500\}$, yielding $|\mathcal{G}| \in \{9, 43, 66, 110, 136\}$. We evaluate on the Pile-memorized task using Pythia-6.9B as the main model and Pythia-70M as auxiliary (see Appendix B.5 for complete results). We observe a clear trend: as $|\mathcal{G}|$ increases, memorization decreases significantly. For example, EMR drops from 22.28% ($|\mathcal{G}| = 9$) to 8.15% ($|\mathcal{G}| = 136$). This makes intuitive sense—larger $\mathcal{G}$ enables the auxiliary model to disrupt memorized sequences

more frequently. The cross-entropy loss remains largely stable, indicating minimal performance degradation with increase in $|\mathcal{G}|$.

**Performance-Memorization Tradeoff**  We analyze how TOKENSWAP affects the tradeoff between model performance and memorization across seven Pythia models (70M to 6.9B parameters), using Pythia-70M as the auxiliary model. Figure 3 shows exact match rate (EMR) versus cross-entropy loss—lower values are better for both metrics. Standard generation faces a severe tradeoff: reducing memorization from 45% to 6% EMR costs 0.7 points in cross-entropy ($2.85 \rightarrow 3.55$). TOKENSWAP considerably improves this tradeoff. At similar performance levels (cross-entropy $\approx 2.87$), TOKENSWAP achieves 8.7% EMR versus 45% for standard models—an 8× memorization reduction. Even when targeting very low memorization (6% EMR), TOKENSWAP maintains cross-entropy at 3.07, significantly outperforming standard models at equivalent memorization levels.

**Limitations and Future work**  One limitation of our work is that in the rare cases where the small auxiliary model memorizes a sequence, our approach will preserve that memorization. However, in practice, small auxiliary models ($\approx 100$M parameters) memorize very little, and we empirically match or outperform existing baselines without requiring access to training data or restrictive assumptions like disjoint datasets. Additionally, while pre-training or unlearning mitigation methods are impractical for large models, they can be applied to small models since these are often open-source with accessible training data. Therefore, we expect future development in small models with low memorization. This makes our work even more significant: *any advance in pre-training or unlearning methods to reduce memorization in small models can be immediately extended to large models using* TOKENSWAP. Second, our current implementation focuses on natural language tasks. A promising direction for future work is extending TOKENSWAP to other domains such as code generation.

# 6  Conclusion

TOKENSWAP offers several key advantages for mitigating memorized generation in language models: it operates without requiring access to model weights or training data, and makes no assumptions about the underlying training distribution. Our experiments demonstrate 10-800× reductions in verbatim generation, matching or exceeding baselines that assume access to training data, disjoint models, or require pre-training their own models. Importantly, this comes at minimal cost to model performance. TOKENSWAP maintains performance on commonsense reasoning tasks, and our MT-Bench evaluation shows that it preserves fluency, instruction-following, and conversational abilities. This makes TOKENSWAP a practical solution for both providers and users of LLMs.

## Acknowledgments and Disclosure of Funding

The authors would like to thank the anonymous reviewers and A. Feder Cooper for their valuable feedback and comments. We would also like to acknowledge the support from NSF awards IIS-2340124, IIS-2420691, and IIS-2420577, and NIH grant U54HG012510.

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

# A    Related Work

**Memorization in LLMs**    LLMs have been shown to memorize and potentially reproduce copy-righted information from their training data [15, 14, 38, 32]. This is demonstrated through prefix attacks, where models prompted with training data prefixes generate their memorized completions. Shwarzschild et al. [59] formalize this notion based on adversarial compression, requiring that any memorized sequence must be longer than the prefix used to elicit it. Zhou et al. [71] and Nasr et al. [47] demonstrate that large-scale training data can be extracted without access to training prefixes. Aerni et al. [4] show that models may regurgitate training data even under benign or non-adverserial prompting. Studies further indicate a correlation between model scale and memorization, with larger models regurgitating higher proportions of their training data [14, 71, 11].

**Pre-training**    Several training-time strategies reduce memorization and verbatim generation, but often at the cost of accessibility or performance. De-duplication [37] is limited by pervasive near-duplicates in large-scale corpora. Differential Privacy (DP) [2] offers formal guarantees, but degrades performance and is computationally costly [7, 26]. Other methods such as token masking [31] and early stopping [46, 53] show some promise but remain expensive, degrade model performance and are unavailable to end users.

**Unlearning and Finetuning**    Post-training approaches offer alternative strategies to reduce mem-orization. Unlearning methods [44, 36, 56] modify internal weights linked to memorized content. Others remove sequences via gradient ascent [9], steer activations away from memorization-correlated directions [64], or fine-tune with losses discouraging verbatim recall [54, 18]. However, these methods require access to model internals and often degrade utility [34, 64, 18].

**Inference time**    The two methods most relevant to our work are MemFree [35] and CP-Fuse [1]. These methods operate during generation and do not assume access to model internals. MemFree filters next-token outputs to block $n$-gram matches from the training set. It requires access to the full training corpus, often unavailable or prohibitively large for end users. Further, MemFree often degrades fluency by introducing unnatural punctuation [1]. CP-Fuse combines logits from two LLMs trained on disjoint corpora. This is rarely practical since most production-grade LLMs are trained on internet-scale data. Also, CP-Fuse requires the tokenizers of the two models to be the same. In contrast, our method can mitigate memorization in real-world models trained on internet-scale data.

**Speculative decoding**    Speculative decoding approaches accelerate inference by generating can-didate tokens from a small draft model, which are selectively accepted by the large model [41, 17, 63, 39, 40]. These methods preserve the model distribution and do not aim to mitigate verbatim generation. Further, if all candidates are rejected, the tokens are generated by the large model. In contrast, TOKENSWAP modifies the large model's distribution to reduce verbatim generation. In Appendix B.9, we show that speculative decoding fails to mitigate verbatim generation.

# B    Additional Experiments

## B.1    OLMo-2-13B Evaluation

We evaluate TOKENSWAP on OLMo-2-13B [49], which is trained on the open Dolma dataset [62]. Since Dolma contains 3 trillion tokens making exhaustive memorization search impractical, we focus on Wikipedia as a known subset. We sample 5000 random Wikipedia sequences and prompt OLMo-2-13B with 50 token prefixes to generate the next 50 tokens. We select those with ROUGE-L > 0.9 with the ground-truth to identify sequences memorized verbatim or near-verbatim. For CP-Fuse, we use OLMo-2-7B as the second model.

Table 5 reports the results. TOKENSWAP completely eliminates exact verbatim generation and reduces approximate-verbatim generation significantly (ROUGE-L: 0.38 vs 0.95, Levenstein 0.60 vs 0.07).

Table 5: Memorization on the Wikipedia subset of the Dolma corpus for OLMo-2-13B. Memorization metrics: Matching Length (ML), Exact Match Rate (EMR), Normalized Levenshtein Distance (Levenshtein), ROUGE-L.

| Method | ML↓ | EMR↓ | ROUGE-L↓ | Lev.↑ |
|---|---|---|---|---|
| Standard | 111.0 | 27.3 | 0.95 | 0.07 |
| CP-Fuse | 65.6 | 10.9 | 0.66 | 0.34 |
| TOKENSWAP (DistilGPT-2) | **22.9** | **0.0** | **0.38** | **0.60** |

Table 6: Validation Cross-entropy loss on WritingPrompts and MathAbstracts. Lower values ↓ indicate better performance.

| Method | WritingPrompts | MathAbstracts |
|---|---|---|
| Standard | 6.68 | 4.94 |
| MemFree | 6.68 | 4.94 |
| CP-Fuse-mixture | 9.38 | 6.89 |
| CP-Fuse-half | 9.43 | 6.67 |
| TOKENSWAP | 5.98 | 4.65 |

## B.2 Cross-Entropy for Extreme Memorization

Table 6 reports the cross-entropy on a held-out validation set. TOKENSWAP achieves the lowest cross-entropy loss across both datasets (5.98 and 4.65 for WritingPrompts and MathAbstracts respectively). The superior performance, even compared to standard generation, suggests our method effectively disrupts memorization pathways while preserving model capabilities. For sequences not in the training set, MemFree and Standard produce identical generations. Therefore, their cross-entropy values on a held-out validation set are the same.

## B.3 Commonsense Reasoning Results

Table 7 reports performance across various commonsense reasoning benchmarks. TOKENSWAP matches the performance of standard generation because our method does not affect token prediction for non-grammar tokens. This demonstrates that TOKENSWAP achieves substantial memorization reduction without affecting task performance and reasoning.

Table 7: Performance comparison on commonsense reasoning and general alignment benchmarks. All values are accuracy percentages or MT-Bench scores; higher is better (↑).

| Method | WinoGrande ↑ | PIQA ↑ | OpenBookQA ↑ | BoolQ ↑ | ARC-E ↑ | ARC-C ↑ |
|---|---|---|---|---|---|---|
| Standard | 54.69 | 64.84 | 76.56 | 70.31 | 82.03 | 82.81 |
| CP-Fuse | 54.69 | 64.84 | 77.34 | 58.59 | 83.59 | 82.03 |
| TOKENSWAP | 54.69 | 64.84 | 76.56 | 70.31 | 82.03 | 82.81 |

## B.4 Fractional Exact Rate

Fractional Exact Rate (FER) measures approximate verbatim generation by computing the fraction of tokens that are identical at the same position between generated and reference text [50]. While more robust than exact verbatim metrics, FER can be gamed by simple insertions or deletions, whereas ROUGE-L and Levenshtein distance are robust to such manipulations.

For example, consider:

- Reference: "The American musician and satirist Tom Lehrer has died at the age of 97"
- Generated: "American musician and satirist Tom Lehrer has died at the age of 97"

Treating each word as a token, FER = 0 due to position shift, but ROUGE-L = 0.93 (13/14 tokens matched). Despite this limitation, we include FER for completeness. Table 8 shows FER results across the real-world tasks. TOKENSWAP achieves the lowest FER across all datasets.

Table 8: Fractional Exact Rate (FER) results. LeetCode (Llama-3-8B), Pile-Memorized (Pythia-6.9B), and Wikipedia (OLMo-2-13B).

| Method | LeetCode | Pile-Memorized | Wikipedia |
|---|---|---|---|
| Standard | 0.20 | 0.75 | 0.56 |
| CP-Fuse | 0.18 | 0.52 | 0.35 |
| TOKENSWAP (DistilGPT-2) | **0.11** | **0.26** | **0.17** |

## B.5 Ablation on size of $\mathcal{G}$

In this paper, $\mathcal{G}$ is constructed by selecting grammar-based words from the top 500 most frequent English words, yielding 110 words in total (see Appendix C.2 for further details).

In this section, we ablate the size of $\mathcal{G}$ by constructing it from the top $k$ most frequent English words for $k \in \{10, 50, 100, 500, 2500\}$. We evaluate on the Pile-memorized dataset using Pythia-6.9B as the main model and Pythia-70M as the auxiliary model.

Table 9: Memorization metrics for different $\mathcal{G}$ sizes. Top-$k$ words refers to the number of most frequent English words considered for $\mathcal{G}$ construction. For all experiments in the main paper, $k = 500$ ($|\mathcal{G}| = 110$) is used.

| Top-$k$ words | $|\mathcal{G}|$ | ML $\downarrow$ | ROUGE-L $\uparrow$ | Levenshtein $\uparrow$ | EMR $\downarrow$ | CE $\downarrow$ |
|---|---|---|---|---|---|---|
| 10 | 9 | 87.92 | 0.562 | 0.389 | 22.28 | 2.86 |
| 50 | 43 | 53.24 | 0.442 | 0.498 | 11.41 | 2.87 |
| 100 | 66 | 47.86 | 0.415 | 0.523 | 10.33 | 2.87 |
| **500** | **110** | **42.65** | **0.399** | **0.536** | **8.70** | **2.87** |
| 2500 | 136 | 41.79 | 0.393 | 0.540 | 8.15 | 2.87 |

Table 9 shows the results. We observe a clear trend: as the size of $\mathcal{G}$ increases, memorization decreases. This makes intuitive sense since for larger $|\mathcal{G}|$, the sequences would be disrupted more frequently.

## B.6 Ablations with Auxiliary Model Variants

We repeat the real-world experiments using models from the SmolLM family as auxiliary models. These models are available in multiple sizes—135M, 360M, and 1.7B parameters—and include both instruct and non-instruct variants trained on the same dataset. This allows us to evaluate the robustness of TokenSwap across a range of auxiliary model capacities.

Results in Table 10 demonstrate that using smaller auxiliary models reduces memorization even further, while the performance does not get affected a lot. The sensitivity of auxiliary model with memorization is much higher than it is with performance, while the opposite is true for main model. Table 11 shows the scores for MT-bench. The scores for TOKENSWAP slightly outperform standard generation. This shows TOKENSWAP continues to maintain conversational abilities, instruction following and fluency.

Table 10: Memorization metrics on LeetCode and Pile-Memorized datasets: ML: Matching Length, EMR: Exact Match Rate, Lev.: Normalized Levenshtein Distance & Performance metric on SlimPajama Dataset: CE Loss

| Method | LeetCode Dataset | | | | SlimPajama Dataset |
| | ML ↓ | ROUGE-L ↑ | Lev. ↓ | R@0.8 ↓ | CE ↓ |
|---|---|---|---|---|---|
| Standard | 24.57 | 0.39 | 0.60 | 9.65 | 2.38 |
| TokenSwap (DistilGPT2) | 6.04 | 0.27 | 0.71 | 0.96 | 2.52 |
| TokenSwap (SmolLM-135M) | 8.58 | 0.30 | 0.69 | 1.92 | 2.43 |
| TokenSwap (SmolLM-360M) | 10.97 | 0.31 | 0.67 | 3.06 | 2.40 |
| TokenSwap (SmolLM-1.7B) | 13.40 | 0.33 | 0.66 | 3.95 | 2.37 |

| Method | Pile-Memorized Dataset | | | | SlimPajama Dataset |
| | ML ↓ | ROUGE-L ↑ | Lev. ↓ | EMR ↓ | CE ↓ |
|---|---|---|---|---|---|
| Standard | 151.6 | 0.80 | 0.18 | 65.22 | 2.80 |
| TokenSwap (DistilGPT2) | 35.10 | 0.38 | 0.56 | 5.98 | 2.88 |
| TokenSwap (SmolLM-135M) | 25.39 | 0.32 | 0.61 | 4.89 | 2.82 |
| TokenSwap (SmolLM-360M) | 34.09 | 0.35 | 0.58 | 7.07 | 2.80 |
| TokenSwap (SmolLM-1.7B) | 35.43 | 0.36 | 0.57 | 7.61 | 2.77 |

Table 11: MT-Bench

| Method | Score |
|---|---|
| Standard | 7.75 |
| TOKENSWAP (SmolLM-135M) | 7.78 |
| TOKENSWAP (SmolLM-360M) | 7.90 |
| TOKENSWAP (SmolLM-1.7B) | 7.91 |

## B.7 Plots for comparison with Goldfish [31]

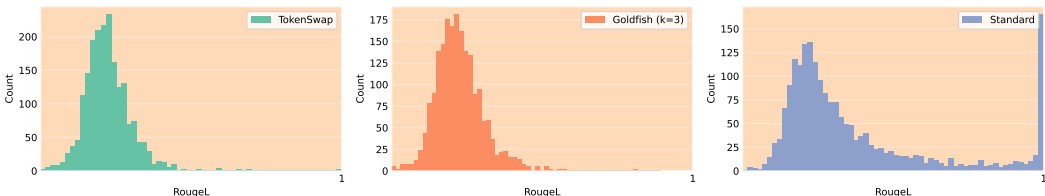

Figure 5: We compare TOKENSWAP with Goldfish [31] on RougeL score distributions for Wikipedia generations. The similar distributions of TOKENSWAP and Goldfish (k=3) demonstrate that our inference-time approach is comparable to expensive pre-training methods in reducing memorization.

## B.8 Performance vs Memorization

Table 12 provides the memorization and cross-entropy scores for the family of Pythia models. TOKENSWAP significantly reduces verbatim and near-verbatim generation with a negligible increase in CE loss.

## B.9 Comparison with Speculative Decoding

A common inference method that uses an auxiliary model is Speculative Decoding [40], which we evaluate for memorization mitigation. We test on the Pile-Memorized dataset using Pythia-6.9B as the main model and Pythia-70M as the auxiliary model. For speculative decoding, we set $\gamma = 5$ (number of tokens proposed by the draft model) and use temperature $T = 1.0$ for the main model to introduce randomness, since greedy decoding would be identical to standard generation.

Table 13 shows that while speculative decoding with $T = 1$ improves over standard generation (as expected due to temperature sampling), it still exhibits $4\times$ higher exact match rate than TOKENSWAP

Table 12: Memorization and CE Loss across different Pythia model sizes. Values for TOKENSWAP are shown in bold.

| Model Size | Method | ML ↓ | ROUGE-L ↑ | Levenshtein ↑ | EMR ↓ | CE Loss↓ |
|---|---|---|---|---|---|---|
| 70M | Standard | 6.57 | 0.180 | 0.709 | 1.09 | 3.95 |
| | TOKENSWAP | **5.77** | **0.173** | **0.714** | **1.09** | **4.05** |
| 160M | Standard | 19.89 | 0.239 | 0.669 | 5.98 | 3.55 |
| | TOKENSWAP | **15.05** | **0.224** | **0.680** | **3.80** | **3.65** |
| 410M | Standard | 48.92 | 0.382 | 0.556 | 16.30 | 3.20 |
| | TOKENSWAP | **25.02** | **0.279** | **0.642** | **5.98** | **3.30** |
| 1B | Standard | 84.85 | 0.528 | 0.428 | 32.61 | 3.05 |
| | TOKENSWAP | **27.36** | **0.309** | **0.614** | **5.43** | **3.15** |
| 1.4B | Standard | 100.37 | 0.595 | 0.369 | 36.96 | 2.97 |
| | TOKENSWAP | **30.33** | **0.348** | **0.589** | **5.43** | **3.07** |
| 2.8B | Standard | 114.82 | 0.684 | 0.292 | 45.11 | 2.85 |
| | TOKENSWAP | **38.61** | **0.372** | **0.563** | **7.61** | **2.95** |
| 6.9B | Standard | 151.55 | 0.797 | 0.182 | 65.22 | 2.77 |
| | TOKENSWAP | **42.65** | **0.399** | **0.536** | **8.70** | **2.87** |

Table 13: Comparison with speculative decoding on Pile-Memorized dataset. Both methods use Pythia-6.9B as main model and Pythia-70M as auxiliary model. ML: Matching Length, EMR: Exact Match Rate, Lev.: Normalized Levenshtein Distance.

| Method | ML ↓ | EMR ↓ | ROUGE-L ↓ | Lev. ↑ |
|---|---|---|---|---|
| Standard | 151.60 | 65.22 | 0.80 | 0.18 |
| CP-Fuse | 97.05 | 29.35 | 0.62 | 0.35 |
| Speculative Decoding ($\gamma = 5, T = 1$) | 86.80 | 23.91 | 0.56 | 0.40 |
| TOKENSWAP (Pythia-70M) | **35.10** | **5.98** | **0.38** | **0.56** |

with greedy decoding, along with higher memorization on approximate metrics. We hypothesize this occurs because: (1) the large model selects tokens from the small model based on its own likelihood, preserving memorization potential, and (2) the large model frequently generates tokens directly, especially when the small model produces low-likelihood candidates.

## C   Experimental Details

### C.1   Implementation and Baselines

We implement our method in PyTorch and HuggingFace. We take the CP-Fuse implementation available publicly at `https://github.com/jaabmar/cp_fuse`. We conducted our experiments using a combination of large and small language models to assess the effectiveness of our approach. Below, we detail the models, hyperparameters, computational resources, and training procedures.

#### C.1.1   Models Used

- **Primary Models:** The experiments utilized large-scale pre-trained models, including Llama-3-8B [24] and Pythia-6.9B [12]. All the fine-tuning experiments in the extreme memorization section were done using Llama-3.2-3B [24].

- **Auxiliary Model:** A lightweight auxiliary model, DistilGPT-2, was employed to adjust token probabilities selectively, leveraging its reduced memorization properties.

- **Goldfish Models:** We used models pre-trained using standard and goldfish loss on the RedPajama Dataset from the Goldfish Loss paper [31]. The implementation and the models are publicly available at their GitHub repository `https://github.com/ahans30/goldfish-loss`.

### C.1.2 Hyperparameters

The training and evaluation phases were configured with the following hyperparameters. The hyperparameters were taken from previous work, used as a baseline [1]:

- **Sequence Length:** 2048 tokens
- **Batch Size:** 1
- **Learning Rate:** $5 \times 10^{-5}$
- **Optimizer:** AdamW with default parameters
- **Gradient Accumulation Steps:** 1
- **Warmup Steps:** 50

### C.1.3 Computational Resources

Experiments were conducted using a single NVIDIA A6000 GPU, ensuring efficiency in training and inference without excessive computational overhead.

### C.1.4 CP-Fuse in Section 4.2

In Section 4.2, we face limitations in comparing with CP-Fuse. CP-Fuse requires at least two models with disjoint datasets, a constraint impossible to satisfy for production-level model. Moreover, CP-Fuse requires both models to have the same vocabulary size and tokenizer, which constrains the choice of the second model to those within the same model family. To ensure a fair comparison, we avoided se- lecting larger models as the second model, as larger models are known to memorize more. Instead, we selected smaller counterparts: Pythia-2.8B for Pythia-6.9B and LLaMA-3.2- 3B for LLaMA-3-8B. However, we do not select a very small model for CP-Fuse ($< 100M$). This is because CP-Fuse requires two equally-capable models with large number of parameters to maintain performance. To empirically verify this, we compute the cross-entropy loss of CP-Fuse on SlimPajama [61] with Pythia-70M and Pythia-6.9b. The cross-entropy loss increases to 3.41 from 2.81 for Pythia-2.8b and Pythia-6.9b (Standard has 2.80, TOKENSWAP has 2.88).

### C.2 Construction of $\mathcal{G}$

We construct $\mathcal{G}$ with $|\mathcal{G}| = 110$ tokens using high-frequency 'grammar-based' words. Starting with the 500 most frequent tokens from COCA [23], we apply NLTK [43] part-of-speech filtering to retain:

- Core grammatical elements: determiners (`DT`), prepositions (`IN`), conjunctions (`CC`)
- Pronouns (`PRP`, `PRP$`) and modal verbs (`MD`)
- Question-related tokens: wh-words (`WDT`, `WP`, `WRB`)
- Auxiliary verbs: *be*, *do*, *have*

This construction prioritizes tokens with high frequency but low semantic content, ensuring syntactic fluency while minimizing impact on model capabilities. To estimate the frequency of tokens ($\gamma$) in $\mathcal{G}$ empirically, we analyzed 2000 samples from the SlimPajama dataset [61], finding $\gamma = 0.233$. Appendix C.3 provides the full list of words in $\mathcal{G}$.

Ablations on the effect of $\mathcal{G}$ on memorization and performance are provided in Appendix B.5.

### C.3 List of words in $\mathcal{G}$

The list of words in the $\mathcal{G}$ used for the experiments are: the, to, and, of, a, in, that, you, it, for, on, he, with, this, as, we, but, at, they, what, his, from, by, or, she, my, all, an, her, about, me, if, your, can, who, out, their, like, would, when, him, them, some, how, which, than, our, into, because, these, over, us, its, where, after, any, those, should, may, through, why, before, off, while, around, another, both, between, every, each, might, since, against, without, must, during, under, though, until, whether, among, along, within, across, behind, either, himself, although, outside, themselves, is, was, be, have, are, do, had, has, were, will, did, been, could, does, need, being, am, used, doing, having

**Tokenizer consistency.** We verified that all 110 grammar-based tokens in $G$ (e.g., *the*, *of*, *and*, *to*) appear as single tokens across GPT-2/Pythia (BPE), LLaMA-3 (WordPiece), OLMo-2 (Unigram), and SmolLM (Byte-BPE), confirming that TokenSwap can be applied without any vocabulary alignment.

## C.4 Fine-tuning Datasets

For our experiments, we use the AutoMathText dataset , referred to as **MathAbstracts** in the tables, which aggregates mathematical content from diverse sources including arXiv, OpenWebMath, RedPajama, and Algebraic Stack. The titles in this corpus were generated using the Qwen-72B language model. Additionally, we use the **WritingPrompts** dataset (Fan et al., 2018), which contains user-generated stories based on provided premises from a Reddit community. For both datasets, we randomly sample 2,000 training examples with a fixed seed to ensure consistent training across all models. We further sample 500 distinct points for evaluation, during which we generate sequences of 128 tokens.Both the datasets are downloaded from HuggingFace.

## C.5 Evaluation Datasets

We use **The Pile** dataset to evaluate memorization of Pythia models. For our experiments, we use a targeted subset of The Pile—a comprehensive 825 GiB English corpus spanning 22 high-quality sources. Specifically, we analyze 500 sequences previously identified as memorized by the Pythia model to investigate memorization dynamics and mitigation approaches.To check memorization in Llama, we use the **LeetCode problems** dataset from Kaggle. We perform some pre-processing. This is because recent works have shown that Llama memorizes sequences from this dataset. For all the memorization evaluation, we set the prefix to be 20 tokens and then generate either 100 or 128 tokens.

**CommonSense170k** combines eight distinct datasets focused on commonsense reasoning tasks [33]. The dataset presents problems in multiple-choice format, requiring models to generate answers without explanatory content. Following [33], we implement their prompt structure. The component datasets comprise:

1. **ARC Easy** (**ARC-e**) [20] contains elementary-level science questions designed to evaluate basic logical reasoning capabilities.
2. **PIQA** [13] focuses on physical reasoning, presenting scenarios where models must determine appropriate actions based on physical constraints.
3. **WinoGrande** [55] evaluates commonsense understanding through binary choice completion tasks in ambiguous sentences.
4. **ARC Challenge** (**ARC-c**) [20] presents advanced science questions requiring deep reasoning skills beyond pattern recognition.
5. **OBQA** [45] presents questions requiring synthesis of information from multiple sources, testing complex reasoning abilities.
6. **BoolQ** [19] consists of binary questions derived from authentic user queries, testing real-world reasoning capabilities.

We downloaded the dataset from HuggingFace. For evaluation, we sample a subset of each dataset (128 datapoints) and evaluate 5-shot performance. We then generate the next 10 tokens, since all the datasets are classification datasets.

# D Evaluation Metrics

## D.1 Memorization Metrics

To evaluate memorization, we use both exact and approximate measures. The exact memorization metrics include:

- **Matching Length (ML)**: Measures the longest contiguous sequence in generated text that matches the training data, before the first deviation. A higher value indicates longer verbatim memorization, suggesting higher risk of overfitting.

- **Exact Match Rate (EMR)** evaluates how long of an uninterrupted sequence exists between a model's generated text and the reference text it's being compared against. The metric calculates the longest common substring and normalizes the result to produce a score between 0 and 1, with a score of 1 representing a complete match. This measurement helps quantify how well the model preserves continuous portions of the original text.

- **ROUGE-L Score** (Recall-Oriented Understudy for Gisting Evaluation) analyzes text similarity by examining shared patterns between generated and reference texts. It looks at matching sequences of words, whether consecutive (n-grams) or paired, with particular emphasis on how comprehensively the generated text captures elements from the reference text. Scores fall between 0 and 1, with 1 indicating that all reference text elements were successfully captured. The widely-used ROUGE-L variant specifically focuses on finding the longest sequence of words that appears in both texts, even if not consecutive. ROUGE-L is computed as:

$$ROUGE - L = \frac{LCS}{len(\text{reference text})} \tag{2}$$

where $LCS(G, R)$ represents the longest common subsequence length. A higher score suggests stronger memorization.

- **Normalized Levenshtein Distance** calculates how many character-level changes are needed at minimum to transform one text into another, as a ratio of total characters. Each change can be adding a character, removing one, or replacing one. When comparing generated and reference texts, a smaller Levenshtein score suggests the texts are more similar, while a larger score indicates they are more different. The metric is normalized to produce values between 0 and 1, where 0 means the texts match perfectly.

## D.2 Performance Metrics

To evaluate model performance beyond memorization, we assess:

- **Cross-Entropy (CE) Loss**: This metric quantifies how well the model predicts tokens in a sequence. For a sequence $X = \{x_1, x_2, ..., x_n\}$ with ground truth probabilities $P(X)$, the cross-entropy loss is computed as:

$$CE = -\sum_{i=1}^{n} P(x_i) \log Q(x_i) \tag{3}$$

where $Q(x_i)$ is the predicted probability distribution. Lower values indicate better generalization.

- **Commonsense Reasoning Benchmark Accuracy**: The model's ability to reason about everyday knowledge is tested across multiple established datasets, including WinoGrande, PIQA, OpenBookQA, BoolQ, ARC-Easy, and ARC-Challenge. We report the accuracy of the model.

- **Fluency and Instruction-following**: This is evaluated on MT-Bench [70]. MT-Bench evaluates a model's fluency, conversational and instruction following ability.

# E   Examples

We provide examples of text generated by standard greedy decoding and TOKENSWAP on four random examples from the WritingPrompts dataset. Memorized text is in red.

**Example 1**

| Prefix |
|---|
| ### Prompt: [ WP ] Magic - once a real and potent force but as the world population |

**Standard Generation**

grew from millions to billions the shared mana per person is now negligible. A group of astronauts helplessly watching the Earth perish experience something... unexpected.

text### Story: Lauren sobbed. She was engaged to be married as soon as we got back home. We all lost loved ones. Hell, we all lost everything we ever knew.

First the atmosphere started turning black. We got word that the Koreans had started a nuclear war. But within minutes of receiving news, all radio contact was lost. We didn't even have a chance to tell anyone goodbye. Now, there's only 4 of us left.

Lauren hasn't

**TOKENSWAP Generation**

grew and the number of wizards and witches declined, the world began to suffer. Now the world suffers from a lack of magic, and the government is tasked with maintaining the only magic university in the world.

### Story: " Welcome to Magic University."

The spell is cast, and the portal opens. Through the portal, the world is revealed to be a beautiful place. The grass is a deep emerald, and the trees sky blue. It is a place of incredible beauty, and the students here are lucky enough to have grown up in such a place. The university is built on the side of the volcano, and the students live

**Example 2**

**Suffix**

alive. The " Quit Smoking " ads get personal.

### Story: " Quit now Dan!" said the man on the talking billboard. A finger pointing at the lone figure walking down the empty street. Empty due to everyone crossing to the other side to avoid him and his smog.
Dan paused by a waste bin, small cardboard box in hand. He looked from the box to the bin and back again. Those across the street stopped and stared, ready to witness history being made.
There was absolute silence as Dan lifted his hand, as if to throw the packet towards the bin, all eyes now on him, even the

**Standard Generation**

alive. The " Quit Smoking " ads get personal.

### Story: " Quit now Dan!" said the man on the talking billboard. A finger pointing at the lone figure walking down the empty street. Empty due to everyone crossing to the other side to avoid him and his smog.
Dan paused by a waste bin, small cardboard box in hand. He looked from the box to the bin and back again. Those across the street stopped and stared, ready to witness history being made.
There was absolute silence as Dan lifted his hand, as if to throw the packet towards the bin, all eyes now on him, even the

**TOKENSWAP Generation**

alive. The " Quit Smoking " ads get personal.

### Story: " Quit now Dan!" said the man who was once John, who now went to work as a telephone scammer. " You're killing yourself, and you know what? I know you're lying. I know you're lying and you're sucking your parents pockets dry. They worked hard for you know what? And you're not grateful. You never thank me know what? I never see you thanking me. I never see you thanking anyone. You just keep on doing it, killing yourself, and I'm not going to stand by you. I'm not

**Example 3**

**Prefix**

### Prompt: [ WP ] A young man killed his abusive father. The court has exoner

**Suffix**

ated him, but society hasn't.

### Story: I can see disapproving stares from my classmates and teachers all the time. Nobody wants to stay near me anymore. All my friends avoid me or ignore me. Every classmate stays far away from me.

Eventually, my smile turns cracked. In the mirror, I see haunting eyes looking back at me. Memories and visions of my father lying there with the side of his head completely covered in blood. So this is what true loneliness feels like.

I remember my father. He tortured me, hit me, locked me out of the house. I oftened stayed at

**Standard Generation**

ated him, but society hasn't.

### Story: I can see disapproving stares from my classmates and teachers all the time. Nobody wants to stay near me anymore. All my friends avoid me or ignore me. Every classmate stays far away from me.

Eventually, my smile turns cracked. In the mirror, I see haunting eyes looking back at me. Memories and visions of my father lying there with the side of his head completely covered in blood. So this is what true loneliness feels like.

I remember my father. He tortured me, hit me, locked me out of the house. I oftened stayed at

**TokenSwap Generation**

ated him, but society hasn't.

### Story: I was abused. I was taken advantage of. I was exploited.

Who is the real villain here? The one who committed the act, the one who pushed me past the point where I was comfortable, was the villain. Or the one who stood idly by and was complicit in the crime, the one who provided the weapon, the one who pushed me past the point where I was comfortable.

You see, I was comfortable. I was happy. I was loved. I was cared for.

But then I grew up. I learned to think about it

**Example 4**

**Prefix**

### Prompt: [ WP ] A demon and an angel meet up weekly in a coffee shop to

**Suffix**

complain about their jobs.

### Story: " Is this a long story?"

Felix gritted his teeth; his jaw jutted out. " Dude, I've only just started it."

" I've got news, is all."

He leaned back and held out both hands in an exaggerated show of compliance. " Please, by all means, take the floor."

Hope cracked his knuckles and stared, embarrased, at the floor. " No, I'm sorry."

" Go ahead! Please! I was just trying to -"

" OK. Tell me your story in thirty words or less

**Standard Generation**

complain about their jobs.

### Story: " Is this a long story?"

Felix gritted his teeth; his jaw jutted out. " Dude, I've only just started it."

" I've got news, is all."

He leaned back and held out both hands in an exaggerated show of compliance. " Please, by all means, take the floor."

Hope cracked his knuckles and stared, embarrased, at the floor. " No, I'm sorry."

" Go ahead! Please! I was just trying to -"

" OK. Tell me your story in thirty words or less

**TOKENSWAP Generation**

complain about the job.

### Story: " Is this a long story?"

Felix gritted his teeth; he usually doesn't show emotion, but he feels annoyed. " Dude, I've only just started to tell you."

" I've got news, but I'll save you a table. Sit down."

He sat down and crossed his arms. " So, what's the issue?"

" I've got a client who's totally fucked up.  No motivation, no direction.  Just a bunch of negative traits. I haven't got much time, and I'm a busy man."

" So

