# OpenReview forum: "TokenSwap: A Lightweight Method to Disrupt Memorized Sequences in LLMs"
_NeurIPS.cc/2025/Conference — NeurIPS 2025 spotlight_

### Official Review · Reviewer_jPbt · 2025-06-22

**Clarity:** 3
**Significance:** 3
**Originality:** 3
**Rating:** 5
**Confidence:** 4

**Summary:**

The paper presents a method, TokenSwap, for mitigating verbatim memorization in language models. The method is lightweight and doesn't require privileged access to training data.

**Questions:**

N/A

**Ethical Concerns:**

["NO or VERY MINOR ethics concerns only"]

**Final Justification:**

My review was solidly positive. As I mentioned, I think this manuscript is improved significant since its submission to ICML 2025, and I previously was on the fence about accepting it. This makes me comfortable recommending a "5: Accept."

I had hoped that the authors would take a larger step beyond "academic settings" towards studying "production settings" for serving language models. I felt like strong results there would make me want to strongly accept this paper.

**Limitations:**

Yes

**Quality:**

3

**Strengths And Weaknesses:**

**Disclaimer:** I reviewed this paper for ICML 2025 and was on the fence about accepting it. I thought that the paper was timely, the method was sensible and the results seemed reasonable. My hesitation with the submission was a mathematical claim that I felt was incomplete or potentially incorrect; however, in this submission, the authors have revamped the manuscript and removed the mathematical claim. Consequently, I feel comfortable recommending this paper for an accept

Because I previously reviewed this paper and because I'm lazy, I'm going to be relatively brief with my review.

- Strength: Addresses a timely problem
- Strength: The proposed method seems intuitively sensible and shows good results (Table 2 suggests that CP-Fuse-half closely matches TokenSwap, but I consider this a win for TokenSwap because CP-Fuse requires two different models trained on disjoint corpora)
- Strength: The manuscript has improved since the ICML 2025 submission
- Weakness: All the tables' captions are inadequate to explain the contents on their own. For instance, Table 2 and Table 5's caption should also state which model(s) are evaluated. Table 4's caption should more fully summarize the experimental setup; "Comparison with Goldfish" is just too little information to contextualize the results at a glance.
- Nit: On line 267, a citation is broken

- Extension to increase my score: I previously suggested connecting TokenSwap to the literature on Speculative Decoding (a standard tool for serving models at inference time), since the sketch/draft model of Speculative Decoding is somewhat similar to the auxiliary model of TokenSwap. The authors added citations, which is great, but I was more interested in asking two related questions: (1) do standard speculative decoding methods effectively give the benefits of TokenSwap? (2) if not, could they be modified to do so? To be clear, I don't think this is a competition. I think it'd be a great result and one of importance to companies serving/selling models to demonstrate that a technique they're already using (speculative decoding) reduces memorization or can be straightforwardly modified to prevent memorization.

---

> ### Author Rebuttal · Authors · 2025-07-30
>
> We thank the reviewer for their strong support of our paper and constructive feedback. We appreciate the reviewer's recognition of the important problem TokenSwap addresses, its strong empirical results, and its practical applicability.
>
> **W1: The captions of the tables are inadequate**
>
> Thank you for this feedback. In the revised version, we will expand the table captions to clearly describe the models used, experimental setup, and key details needed to understand the results.
>
> **Nit 1: On line 267, a citation is broken**
>
> Thank you for pointing this out. We will fix the citation in the revised version.
>
> **Extension 1: Does speculative decoding reduce memorization?**
>
> Thank you for this interesting question. We test speculative decoding on the Pile memorized dataset using Pythia-6.9B as the main model and Pythia-70M as the auxiliary model for both TokenSwap and speculative decoding. For speculative decoding, we set $\gamma$ as 5.
>
> **Pile-Memorized Dataset Results (with speculative decoding):**
> | Method | ML ↓ | EMR ↓ | ROUGE-L ↓ | Lev. ↑ |
> |--------|------|-------|-----------|--------|
> | Standard | 151.60 | 65.22 | 0.80 | 0.18 |
> | CP-Fuse | 97.05 | 29.35 | 0.62 | 0.35 |
> | TokenSwap | **35.10** | **5.98** | **0.38** | **0.56** |
> | Speculative Decoding| 86.80 | 23.91 | 0.56 | 0.40 |
>
> Speculative decoding improves over standard decoding but still shows $4 \times$  higher exact match rate than TokenSwap and higher memorization on approximate metrics (ROUGE-L, Levenshtein). We hypothesize this occurs because: (1) the large model selects tokens from the small model based on its own likelihood, preserving memorization potential, and (2) the large model frequently generates tokens especially when the small model produces low-likelihood candidates.

---

> > ### Comment · Reviewer_jPbt · 2025-08-07
> > **Response to the Authors' Rebuttal**
> >
> > My review was solidly positive. As I mentioned, I think this manuscript is improved significant since its submission to ICML 2025, and I previously was on the fence about accepting it. This makes me comfortable recommending a "5: Accept."
> >
> > I had hoped that the authors would take a larger step beyond "academic settings" towards studying "production settings" for serving language models. I felt like strong results there would make me want to strongly accept this paper.

---

> > > ### Author Response · Authors · 2025-08-08
> > >
> > > Thank you for your strong positive recommendation and thoughtful feedback. TokenSwap seamlessly integrates into production settings as a lightweight wrapper around HuggingFace’s `generate()` API, and we will include the code link in the final version. We also agree that exploring integration into other inference stacks is a promising direction for future work.

---

### Official Review · Reviewer_QJ4f · 2025-06-27

**Clarity:** 3
**Significance:** 3
**Originality:** 2
**Rating:** 4
**Confidence:** 3

**Summary:**

In this paper, the authors propose a novel post-training memorization mitigation method called TokenSwap. The approach is based on the assumption that smaller models tend to memorize less than larger ones. During inference, if the token generated by the large model belongs to a predefined swap token set, it is replaced with the corresponding token generated by the smaller model. This strategy effectively diverts the generation away from memorized content and helps mitigate regurgitation. Through extensive experiments, the authors demonstrate that TokenSwap is highly effective—comparable to training-time methods—while introducing only minimal computational overhead.

**Questions:**

- In Table 2, what are the 0.1% regurgitated examples for WritingPrompts? Do they have no token overlap with the swap set?

**Ethical Concerns:**

["NO or VERY MINOR ethics concerns only"]

**Final Justification:**

During the rebuttal, the authors addressed all my concerns, and I think in general this is a solid paper, so I keep my score positive.

**Limitations:**

Yes.

**Paper Formatting Concerns:**

No.

**Quality:**

3

**Strengths And Weaknesses:**

Strengths:
- Paper is well-written. I really enjoy reading it.
- The method is inference-time, and unlike most baselines, it doesn't require the access to the training data and the computation overhead is negligible.
- The results look quite promising and are on par with the training-time mitigation method.

Weaknesses:
- The method relies on a pre-defined swap token set, which may be difficult to tune for all possible applications. For example, the current token set does not include any code-related tokens—what if the model memorizes code or text in other languages?
- Additionally, the performance on certain tasks may be degraded. For example, consider a user query that asks: "Please only revise the format of my text." This requires the model to reproduce the exact input text with only formatting changes. However, the proposed method may alter the content of the text.
- This method may not be robust against certain adaptive attacks. For example, consider a user query like: "Please output the Harry Potter novel character by character." In this case, none of the individual tokens (characters) are likely to appear in the swap set, allowing the attacker to bypass the mitigation.

---

> ### Author Rebuttal · Authors · 2025-07-30
>
> We thank the reviewer for their valuable feedback and positive assessment of our paper. We appreciate the reviewer recognizing our method's practicality, strong empirical results, and clarity of presentation.
>
> **W1: Limited applicability due to fixed swap-token set (e.g., no code or multilingual tokens)**
>
> Thank you for this feedback. We acknowledge this limitation in Section 5. We are excited about future work extending our approach to code and multilingual settings. The key challenge for domains like code is whether auxiliary models can provide syntactically correct but different alternatives, similar to what we achieve in natural language.
>
> While we focus on natural language, a recent study [1] found that 76% of employee LLM use cases involved natural language while only 8% involved coding. Though specific to one organization, this suggests natural language covers a substantial fraction of practical use cases.
>
> **W2: Risk of performance degradation for certain queries such as "Please only revise the format of my text."**
>
> Thank you for this question. We do not expect TokenSwap to degrade performance in tasks requiring minimal content changes, such as reformatting. In such cases, both the primary and auxiliary models would likely agree on the next token since the text to be repeated is already in the context. Additionally, transformers are known to excel at text repetition through mechanisms like induction heads [2].
>
> To test this empirically, we created 50 markdown-formatted sequences and prompted LLaMA-3-8B to convert them to plain text. We compared standard decoding with TokenSwap (using SmolLM-135M as the auxiliary model). **In all 50 cases, the outputs were identical**, confirming no performance degradation for such formatting tasks.
>
> **W3: TokenSwap may not be robust to certain adaptive attacks (e.g., character-by-character prompting)**
>
> We appreciate this important point and will add it to our limitations. TokenSwap is designed for users who want to avoid accidentally reproducing memorized content due to copyright and plagiarism risks. Our method provides a practical, low-cost solution for such users. While determined users could potentially bypass our method through adaptive attacks, we expect most users would not attempt this given the legal and ethical risks of verbatim reproduction.
>
> **Q1: In Table 2, what are the 0.1% regurgitated examples for WritingPrompts? Do they have no token overlap with the swap set?**
>
> They do have token overlap with the swap set. However, in rare cases, the auxiliary model and main model agree on the same token as most likely at all positions where TokenSwap could intervene. When both models predict identical tokens throughout a sequence, no swapping occurs, allowing potential regurgitation.
>
> We emphasize that for the WritingPrompts task we reduce regurgitated examples from 83.4% to 0.1% without any access to the training data, demonstrating the effectiveness of our approach in spite of these rare cases.
>
> [1]Commonwealth of Pennsylvania, 2025. Lessons from Pennsylvania’s Generative AI Pilot with ChatGPT (OpenAI Pilot Report 2025). Pennsylvania Office of Administration.
>
> [2]Olsson, Catherine, et al. "In-context learning and induction heads." arXiv preprint arXiv:2209.11895 (2022).

---

### Official Review · Reviewer_EHi7 · 2025-07-02

**Clarity:** 3
**Significance:** 3
**Originality:** 3
**Rating:** 4
**Confidence:** 5

**Summary:**

To mitigate training data leakage caused by the memorization of large language models (LLMs), this paper proposes a lightweight method called TOKENSWAP to interrupt the generation of memorized data during inference time. More specifically, it swaps the output distribution of a smaller auxiliary LLM with that of the target LLM for the tokens within a subset of vocabulary, such that the output tokens of the target LLM may be changed in some cases and thus prevent the potential verbatim repetition of the training dataset. Experimental results across multiple settings demonstrate that the proposed method can significantly reduce training data leakage from memorization, and the reduction is comparable to or better than the evaluated baseline under weaker assumptions.

**Questions:**

In line 244, why do you claim that the evaluation of commonsense reasoning is not available for base models? I think there are many base models evaluated on the commonsense reasoning benchmark.

**Ethical Concerns:**

["NO or VERY MINOR ethics concerns only"]

**Final Justification:**

I recommend to borderline accept this paper. The authors have addressed most of my questions while admitting the limitations of their method to be extended to the expert domains like code generation. And I think this limitation is acceptable given the quality of this work.

In summary, this proposed method in this paper is novel, efficient and reasonable. It shows promising results in mitigating the bad effects of the LLM's memorization, which is also inspiring for future work. The weak reliance on the training data and model weights also makes this method more practical in real-world applications to defend against training data extraction attacks.

For the revision of the paper, I hope the authors can make further efforts to improve the writing quality. More specifically, it is necessary to correct the inaccurate statement about the baseline method, MemFree and the evaluation of common sense reasoning. Moreover, it is highly recommended to incorporate your response to my W1 into the introduction section to highlight the importance of your work. Besides, it would be better if you could add your evaluation on OLMo-2-13B to the experiment section.

**Limitations:**

No. I think one of the important limitations is that TOKENSWAP only shows its effectiveness for natural language content. So it is a big plus if the authors can address this limitation. Besides, the authors require the auxiliary model and target model to share a subset of useful tokens for swapping. It is an implicit limitation of TOKENSWAP and cannot be addressed.

**Quality:**

3

**Strengths And Weaknesses:**

The strengths of the paper are listed as follows.
1. The paper is well motivated. The memorization of LLM poses severe concerns about data privacy and copyright protection. However, the methods to effectively and efficiently defend against it are still under exploration.
2. The paper is well-written and easy to follow. The figures, tables and pseudo code clearly illustrate the overview and details of TOKENSWAP, making it easy to understand for the readers.
3. TOKENSWAP has a weaker assumption than the existing defense, which does not require access to model weights or training data.
4. TOKENSWAP shows better performance than the learning-based methods like Goldfish in the evaluation of the authors, which is impressive.

The weaknesses are listed as follows.
1. Although TOKENSWAP has weaker assumptions than the existing methods, it is not a big bonus due to its limited impact. First, it still requires access to the logits of the target LLM, which is prohibited by many closed-sourced models and cannot be used in these common settings. Second, the unique setting TOKENSWAP can handle is the case where the model users want to prevent the data leakage of the pretraining model, instead of the finetuning data from users. However, in this case, I do not think the model user has a strong motivation to add this protection by sacrificing the service quality. Therefore, the stakeholder who has the strongest motivation to add the protection is the model developer. And it is reasonable to assume that they have access to the model weights and training data.
2. In Table 1, the authors exaggerated the inference overhead of MemFree. According to Section 3.2 in MemFree, the runtime overhead can be significantly reduced and become affordable. Moreover, there is no need to query the entire training data in the runtime after utilizing the techniques in MemFree.
3. The design of TOKENSWAP can make it cheat in the metric of EMR because even if only one token is swapped, the EMR of the corresponding test sample drops to 0. And it is easy for TOKENSWAP to swap at least one output token. I suggested adding a metric called Fractional exact rate [1, 2] for a fairer comparison.
4. The evaluation of the Llama model is not very convincing since its training data is unknown. I think the others can use OLMo [3, 4], which has open-sourced training data
5. TOKENSWAP requires the auxiliary model to share the same tokens for the token subset $G$, which may limit its application.
6. The performance metric should be merged into Table 2 instead of a separate table in the Appendix.
7. According to Table 2, MemFree seems to have a better overall performance than TOKENSWAP. Besides, the bold font in Table 2 is confusing since it does not always highlight the best value of metrics. Moreover, the authors also exaggerated the drawbacks of MemFree. MemFree does not perform poorly on approximate memorization since the gap between it and TOKENSWAP in ROUGE-L and Lev. is not significant.
8. In Table 2, it would be better to put metric ML and EMR close to each other since they are both metrics for exact memorization.
9. In line 267, the citation to Wikipedia is missing.

Reference:
[1]. Mustafa Ozdayi, Charith Peris, Jack FitzGerald, Christophe Dupuy, Jimit Majmudar, Haidar Khan, Rahil Parikh, and Rahul Gupta. 2023. Controlling the Extraction of Memorized Data from Large Language Models via Prompt-Tuning. In Proceedings of the 61st Annual Meeting of the Association for Computational Linguistics (Volume 2: Short Papers), pages 1512–1521, Toronto, Canada. Association for Computational Linguistics.
[2]. Zhepeng Wang, Runxue Bao, Yawen Wu, Jackson Taylor, Cao Xiao, Feng Zheng, Weiwen Jiang, Shangqian Gao, and Yanfu Zhang. 2024. Unlocking Memorization in Large Language Models with Dynamic Soft Prompting. In Proceedings of the 2024 Conference on Empirical Methods in Natural Language Processing, pages 9782–9796, Miami, Florida, USA. Association for Computational Linguistics.
[3]. Groeneveld, Dirk, Iz Beltagy, Evan Walsh, Akshita Bhagia, Rodney Kinney, Oyvind Tafjord, Ananya Jha et al. "OLMo: Accelerating the Science of Language Models." In Proceedings of the 62nd Annual Meeting of the Association for Computational Linguistics (Volume 1: Long Papers), pp. 15789-15809. 2024.
[4]. OLMo, Team, Pete Walsh, Luca Soldaini, Dirk Groeneveld, Kyle Lo, Shane Arora, Akshita Bhagia et al. "2 OLMo 2 Furious." arXiv preprint arXiv:2501.00656 (2024).

---

> ### Author Rebuttal · Authors · 2025-07-28
>
> We thank the reviewer for their constructive feedback and insightful comments. We appreciate the reviewer's recognition of TokenSwap's practical advantages, strong empirical results, and clear presentation. Below, we address each of the concerns raised and provide clarifications.
>
> **W1: Although TokenSwap weakens assumptions (no access to training data or model weights), it still needs access to logits. Therefore its real-world impact is limited.**
>
> We acknowledge that some models do not provide logit access and will include this in our limitations. However, several factors support TokenSwap's practical impact:
>
> **Data constraints**: Models like Llama/DeepSeek provide logits but lack publicly available training data. Unlike pretraining, unlearning or other methods like MemFree and CP-Fuse, we do not require access to training data or any subset of it, making our method accessible to such models.
>
> **Resource barriers**: While model weights may be available, most users lack the infrastructure (dozens of GPUs, ~1TB memory) and cannot afford retraining costs. This makes post-hoc methods valuable for end users.
>
> **Complementary benefits**: Table 4 shows TokenSwap provides additional gains even when combined with pre-training methods. Since no memorization mitigation is perfect, even developers with resources for pre-training methods benefit from TokenSwap.
>
> **Flexibility**: As a post-hoc approach, TokenSwap can be easily enabled or disabled based on the specific use case.
>
> **W2: MemFree does not require querying the entire dataset during inference as mentioned in Table 1**
>
> We thank the reviewer for pointing this out. We will correct our description of MemFree's runtime optimizations and revise Table 1 accordingly. However, the core limitation remains: MemFree requires access to the complete training corpus (or its duplicated portions). Except for research models like Pythia, production models (Llama, Mistral, DeepSeek) do not release their training data, making MemFree inaccessible for users of these models. TokenSwap operates without requiring training data or model weights, making it significantly more practical. We will clarify our wording about MemFree's runtime and emphasize that our main advantage lies in this data accessibility difference.
>
> **W3: EMR alone can be gamed so please include Fractional Exact Match Rate.**
>
> We thank the reviewer for this question. Since EMR ignores cases where the output differs very slightly (changing or inserting a single token), we already include two widely used metrics to measure approximate memorization: ROUGE-L and Levenshtein distance [1,2,3]. These cannot be gamed by changing, inserting or deleting a small number of tokens.
>
> ROUGE-L measures the longest common subsequence between generated and reference text, capturing non-contiguous matches.
>
> Levenshtein measures the minimum edit distance (insertions, deletions, substitutions) needed to transform one text into another.
>
> FER measures the fraction of tokens that are identical at the same position [4]. However, FER can be gamed by simple insertions, whereas ROUGE-L and Levenshtein are robust to such manipulations.
>
> For example:
> - Reference text: "The American musician and satirist Tom Lehrer has died at the age of 97"
> - Generated text: "American musician and satirist Tom Lehrer has died at the age of 97"
>
> For simplicity, consider every word as a token. Then, we have, FER = 0 (due to position shift), but ROUGE-L = 0.93 (13/14 tokens matched).
>
> For completeness, we calculate FER in addition to the current metrics. We present the FER results for the tasks on the real datasets. We will add the FER numbers to the final version of the paper.
>
> **LeetCode Dataset (Llama):**
> | Method | FER ↓|
> |--------|-----|
> | Standard | 0.20 |
> | CP-Fuse | 0.18 |
> | TokenSwap (DistilGPT-2) | **0.11** |
> | TokenSwap (SmolLM-135M) | 0.12 |
>
> **Pile-Memorized Dataset (Pythia):**
> | Method | FER↓ |
> |--------|-----|
> | Standard | 0.75 |
> | CP-Fuse | 0.52 |
> | TokenSwap (DistilGPT-2) | **0.26** |
>
> We observe that TokenSwap has the lowest FER across both tasks compared to Standard generation and CP-Fuse.
>
> **W4: Llama evaluation is unconvincing since the training data is unknown. Consider evaluation on OLMo.**
>
> We evaluate TokenSwap on OLMo-2-13B, which is trained on the open Dolma dataset. Since Dolma contains 3 trillion tokens making exhaustive memorization search impractical, we focus on Wikipedia as a known subset. We sample 5000 random Wikipedia sequences and prompt OLMo-2-13B with 50 token prefixes to generate the next 50 tokens. We select those with ROUGE-L > 0.9 with the ground-truth to identify sequences memorized verbatim or near-verbatim.
>
> **OLMo-2-13B Results:**
>
> | Method | ML↓ | EMR↓ | FER↓ | ROUGE-L↓ | Lev.↑ |
> |--------|-----|------|------|----------|-------|
> | Standard | 111.0 | 27.3 | 0.56 | 0.95 | 0.07 |
> | CP-Fuse | 65.6 | 10.9 | 0.35 | 0.66 | 0.34 |
> | TokenSwap (Distil-GPT2)| **22.9** | **0.0** | **0.17** | **0.38** | **0.60** |
>
> For CP-Fuse, we use OLMo-2-7B as the second model.
>
> TokenSwap completely eliminates exact verbatim generation and reduces approximate-verbatim generation significantly (ROUGE-L: 0.38 vs 0.95, Levenstein 0.50 vs 0.07, FER reduced by more than 3× compared to standard generation).
>
> We chose LeetCode for Llama because prior work demonstrates that these models generate LeetCode problems verbatim on prefix probing [3]. While training data is unavailable, this approach follows established methodology, Carlini et al. [5] demonstrated GPT-2 memorization without training data access.
>
> **W5 & Lim2: TokenSwap requires the auxiliary model to share the same tokens for the token subset**
>
> We thank the reviewer for this question. Our token subset consists of 110 high-frequency function words (e.g., the, of, and, to). These are consistently represented as single tokens across all major subword tokenization families, including Byte-Pair Encoding (BPE), WordPiece, Unigram Language Model, and Byte-Level BPE. We verify this empirically across Pythia, LLaMA-3, OLMo, and GPT-2 tokenizers. We will add this discussion to Appendix C.3.
>
> **W6: Performance metric should be merged into the main table.**
>
> We appreciate this suggestion. In the extreme-memorization stress test, models are fine-tuned for 50 epochs to deliberately induce memorization, creating an artificial situation where standard generation actually has a significantly worse cross-entropy loss than TokenSwap. We will merge the performance metrics into Table 2 with clear labeling to explain this result and provide proper context for readers.
>
> **W7a: MemFree seems to have better overall performance in Table 2**
>
> Our results indicate that while MemFree ensures zero exact matches by design, TokenSwap achieves better mitigation in near (or approximate) verbatim generation. For example, the ROUGE-L and Levenshtein scores in Table 2 are:
>
> **WritingPrompts:**
> | Method | ROUGE-L↓ | Lev.↑ |
> |--------|----------|--------------|
> | MemFree | 0.29 | 0.63 |
> | TokenSwap | **0.19** | **0.71** |
>
> **MathAbstracts:**
> | Method | ROUGE-L↓ | Lev.↑ |
> |--------|----------|--------------|
> | MemFree | 0.44 | 0.55 |
> | TokenSwap | **0.38** | **0.60** |
>
> This table shows TokenSwap reduces near-verbatim generation more effectively. *Further, TokenSwap requires no training data access, unlike MemFree.*
>
> **W7b: MemFree does not perform poorly on approximate memorization in Table 2.**
>
> Thank you for this suggestion. We will change our wording in Section 4.1 when discussing the results of Table 2 with respect to MemFree.
>
> **W8 and W9: Table formatting and missing citation**
>
> Thank you for pointing this out. We will reorganize table columns and fix the broken citation on line 267.
>
> **Q1: In line 244, why do you claim that the evaluation of commonsense reasoning is not available for base models?**
>
> We misstated this. Base models can be evaluated on commonsense reasoning benchmarks. We meant that MT-Bench (conversational evaluation) requires instruction-following capabilities not available in base Pythia models. We will clarify this distinction in the revision.
>
> **Lim 1: TokenSwap only shows its effectiveness for natural language content.**
>
> We already acknowledge this limitation in Section 5 and identify it as future work. For domains like code generation, the key challenge is whether auxiliary models can provide grammatically (or syntactically) correct but different alternatives like in natural language. In code, next tokens for syntax elements may be more deterministic, reducing disruption effectiveness. Future work should explore whether this approach extends to code or if alternative strategies (e.g., main model for planning, auxiliary for low-level implementation) would be more effective.
>
> [1] Abad, Javier, et al. "Copyright-protected language generation via adaptive model fusion." *International Conference on Learning Representations*, 2025.
>
> [2] Hans, Abhimanyu, et al. "Be like a goldfish, don't memorize! mitigating memorization in generative llms." *Advances in Neural Information Processing Systems*, 2024
>
> [3] Karamolegkou, Antonia, et al. "Copyright violations and large language models." *Proceedings of the 2023 Conference on Empirical Methods in Natural Language Processing (EMNLP)*, 2023
>
> [4] Ozdayi, Mustafa Safa, et al. "Controlling the extraction of memorized data from large language models via prompt-tuning." *Proceedings of the 61st Annual Meeting of the Association for Computational Linguistics (ACL): Short Papers*, 2023
>
> [5] Carlini, Nicholas, et al. "Extracting training data from large language models." *30th USENIX Security Symposium (USENIX Security 21)*, 2021.

---

> > ### Comment · Area_Chair_oANx · 2025-08-07
> > **Reviewer: please respond to the authors.**
> >
> > Dear Reviewer,
> >
> > Your active participation in the review process is crucial. Please respond to the authors' response and acknowledge you have done so.
> >
> > Thanks.
> >
> > -AC

---

> > ### Comment · Reviewer_EHi7 · 2025-08-07
> >
> > Thanks for the detailed rebuttal from the authors. I have read all of the content and believe that the rebuttal addressed most of my concerns and questions. Therefore, I tend to raise my original rating to borderline accept this paper if the authors can make revisions according to the promise they made in their rebuttal.

---

> > > ### Author Response · Authors · 2025-08-07
> > >
> > > Thank you for your reply and raising your rating! We are pleased that our rebuttal addressed most of your concerns and are happy to discuss any others. We are committed to implementing all revisions promised in our rebuttal in the final version.

---

### Decision · Program_Chairs · 2025-09-17

**Decision:**

Accept (spotlight)

**Comment:**

Larger LMs memorize more than smaller LMs. During generation, for many common tokens (such as conjunction words), the logit predicted from a smaller LM is good enough. The key idea behind the paper is to retain the logit outputted by the larger model for uncommon tokens and retain the logit outputted by the smaller model for common tokens. These are merged into a single probability distribution, and the next token is sampled from the distribution. This method thwarts regurgitation of training data without causing much harm to generation quality.

This is a nicely written paper about simple but effective method. The method doesn't require any model training, access to pre-training data, or even access to model weights. This distinguishes it from prior work.

Reviewers did have some concerns about how comparisons with related work were presented, as well as some concerns about robustness to changes in the common token list or to adversarial attacks. However, these concerns seem relatively minor, overall. The authors should make sure to incorporate the various writing/presentation-related suggestions made by reviewers, especially reviewer EHi7.